# Identification of PPARG as key gene to link coronary atherosclerosis disease and rheumatoid arthritis via microarray data analysis

Zhenzhen Zhang[1,2], Yupeng Chen[1,2], Xiaodan Fu[1,2], Linying Chen[1,2], Junlan Wang[1,2], Qingqiang Zheng[1,2], Sheng Zhang[1,2]*, Xia Zhu[3]*

1 Department of Pathology, The First Affiliated Hospital of Fujian Medical University, Fuzhou, Fujian, China, 2 Department of Pathology, National Regional Medical Center, Binhai Campus of the First Affiliated Hospital, Fujian Medical University, Fuzhou, China, 3 Department of Bone Tumor, The Affiliated Hospital of Fujian Medical University, Fuzhou, Fujian, China

* zhgshg@fjmue.edu.cn (SZ); zhuhouy5399@fjmu.edu.cn (XZ)

## Abstract

### Background

Inflammation is the common pathogenesis of coronary atherosclerosis disease (CAD) and rheumatoid arthritis (RA). Although it is established that RA increases the risk of CAD, the underlining mechanism remained indefinite. This study seeks to explore the molecular mechanisms of RA linked CAD and identify potential target gene for early prediction of CAD in RA patients.

### Materials and methods

The study utilized five raw datasets: GSE55235, GSE55457, GSE12021 for RA patients, and GSE42148 and GSE20680 for CAD patients. Gene Set Enrichment Analysis (GSEA) was used to investigate common signaling pathways associated with RA and CAD. Then, weighted gene co-expression network analysis (WGCNA) was performed on RA and CAD training datasets to identify gene modules related to single-sample GSEA (ssGSEA) scores. Overlapping module genes and differentially expressed genes (DEGs) were considered as co-susceptible genes for both diseases. Three hub genes were screened using a protein-protein interaction (PPI) network analysis via Cytoscape plug-ins. The signaling pathways, immune infiltration, and transcription factors associated with these hub genes were analyzed to explore the underlying mechanism connecting both diseases. Immunohistochemistry and qRT-PCR were conducted to validate the expression of the key candidate gene, PPARG, in macrophages of synovial tissue and arterial walls from RA and CAD patients.

### Results

The study found that Fc-gamma receptor-mediated endocytosis is a common signaling pathway for both RA and CAD. A total of 25 genes were screened by WGCNA and DEGs,

**Data Availability Statement:** All relevant data are within the manuscript and its Supporting Information files.

**Funding:** This study was funded by Scientific Research Project of National Key clinical specialty construction project, Grant number (2022YBL-ZD-06); Innovative Medicine Subject of Fujian Provincial Health and Family Planning Commission, China (2022CXA022). The funders had no role in study design, data collection and analysis, decision to publish, or preparation of the manuscript.

**Competing interests:** The authors have declared that no competing interests exist.

which are involved in inflammation-related ligand-receptor interactions, cytoskeleton, and endocytosis signaling pathways. The principal component analysis(PCA) and support vector machine (SVM) and receiver-operator characteristic (ROC) analysis demonstrate that 25 DEGs can effectively distinguish RA and CAD groups from normal groups. Three hub genes TUBB2A, FKBP5, and PPARG were further identified by the Cytoscape software. Both FKBP5 and PPARG were downregulated in synovial tissue of RA and upregulated in the peripheral blood of CAD patients and differential mRNAexpreesion between normal and disease groups in both diseases were validated by qRT-PCR.Association of PPARG with monocyte was demonstrated across both training and validation datasets in CAD. PPARG expression is observed in control synovial epithelial cells and foamy macrophages of arterial walls, but was decreased in synovial epithelium of RA patients. Its expression in foamy macrophages of atherosclerotic vascular walls exhibits a positive correlation (r = 0.6276, p = 0.0002) with CD68.

## Conclusion

Our findings suggest that PPARG may serve as a potentially predictive marker for CAD in RA patients, which provides new insights into the molecular mechanism underling RA linked CAD.

## 1 Introduction

Rheumatoid arthritis (RA) has been shown to be associated with an increased risk of coronary atherosclerosis disease (CAD), which is the buildup of plaque in the coronary arteries due to inflammation, a common characteristic of both RA and CAD [1]. Additionally, RA patients are often treated with medications that can increase their risk of cardiovascular disease [2]. It has been reported that RA Patients have 68% increased risk of developing a myocardial infarction (MI) [3]. However, angina associated MI is insidious due to arthralgia in RA patients, leading to a delay in diagnosis and management [3].

Inflammation is a common pathogenic mechanism for RA and CAD [4]. Chronic inflammation associated with RA can lead to oxidative stress and vascular endothelial injury and dysfunction [5]. Additionally, endothelial dysfunction can cause the release of cytokines such as TNF-α, IL-1, and IL-6 into the systemic circulation in RA patients, which can further promote inflammation and contribute to thrombosis in the coronary artery [6]. However, the inflammatory-mediated biological pathways and molecular mechanisms underlying RA associated CAD are still far from being elucidated. Therefore, it is important to monitor vascular function to reduce risk of heart disease in RA patients.

RA is intrinsically linked to the onset of CAD. However, the current models for predicting CAD linked with RA are hindered by several limitations. First, these models may be restricted by their predictive efficacy and generalization due to insufficient sample sizes of RA linked CAD [7]. Second, inconsistencies exist across scoring systems based on clinical indicators only, which may present more confounding variables in co-existing medical conditions like RA linked CAD [8]. The advent of bioinformatics and high-throughput sequencing technologies has markedly increased our ability to identify key candidate targets and evaluate their predictive efficacy for RA linked CAD.

Potential targets and regulatory biological pathways were identified in RA and CAD datasets, respectively, using microarray bioinformatics technology of recent years. However, there is a dearth of studies that investigate macrophage-related inflammatory processes between RA and CAD. This research aims to fill this gap by employing GSEA analysis to identify common signaling pathways in RA and CAD datasets. WGCNA and ssGSEA were used to identify module genes linked to Fc-gamma receptor phagocytosis. Three hub genes were extracted using STRING and Cytoscape. Then, their predictive abilities were assessed in both diseases through ROC analysis. GeneMANIA and the Cibersort algorithm provided insights into immune infiltration and transcription factors associated with the hub genes. Notably, the involvement of PPARG and macrophages in RA and CAD will be highlighted. Understanding these processes offers new insights into the pathogenesis shared by both diseases and may potentially aid in the development of targeted therapies.

## 2 Materials and methods

### Microarray data download and processing

Fig 1 depicts the study flowchart. Three raw datasets [GSE55235 (n = 20), GSE55457 (n = 23), and GSE12021 (n = 21)] including gene expression data from RA patients and two CAD datasets [GSE42148 (n = 24) and GSE20680 (n = 195)] with healthy controls were downloaded from the GEO database (https://www.ncbi.nlm.nih.gov/geo/). The probe ID was converted into a gene symbol. Gene probes that did not match any gene symbol were removed. Data integration of GSE55235 and GSE55457 was performed by the R package (1.14.0) "InSilicoMerging"[9]. Batch effect removal was performed using the ComBat algorithm [10]. The final matrix merge_exp.txt was used as the training set of RA, while GSE12021 was used as the validation set. As for CAD, GSE42148 was used for the training set and GSE20680 as the validation set.

### GSEA analysis

To identify the common signaling pathways activated in both RA and CAD, the pathogenic and normal samples in both diseases were analyzed using GSEA (v3.15) [11], and the KEGG datasets in the MSigDB database (http://software.broadinstitute.org/gsea/msigdb/index.jsp) were downloaded [12]. The significant enrichment threshold was set to $p < 0.05$ in both diseases. Two common significant pathways were found between the two diseases. Based on the ssGSEA algorithm, the enrichment score of the two intersection pathways was calculated and plotted.

### Differential expression gene analysis

The training sets of both diseases were used to screen DEGs Differential expression analysis was done using a linear model and the empirical Bayes method of the limma package (v 3.10.3)[13]. The differential expression threshold was set as follows: $p<0.05 \& |log2FC|>0.585$.

### WGCNA analysis

Weighted gene co-expression network analysis (WGCNA) uses a hierarchical clustering approach to identify gene modules from the co-expression network. WGCNA measures intra-modular gene connectivity, and highly connected genes are defined as hub genes to screen for clinical feature-associated genes. To screen for co-susceptibility genes from the intersection of WGCNA module genes in RA and CAD, The enrichment score of Fc_gamma R mediated phagocytosis pathway was used as the phenotype, along with the DEGs of the CAD and RA

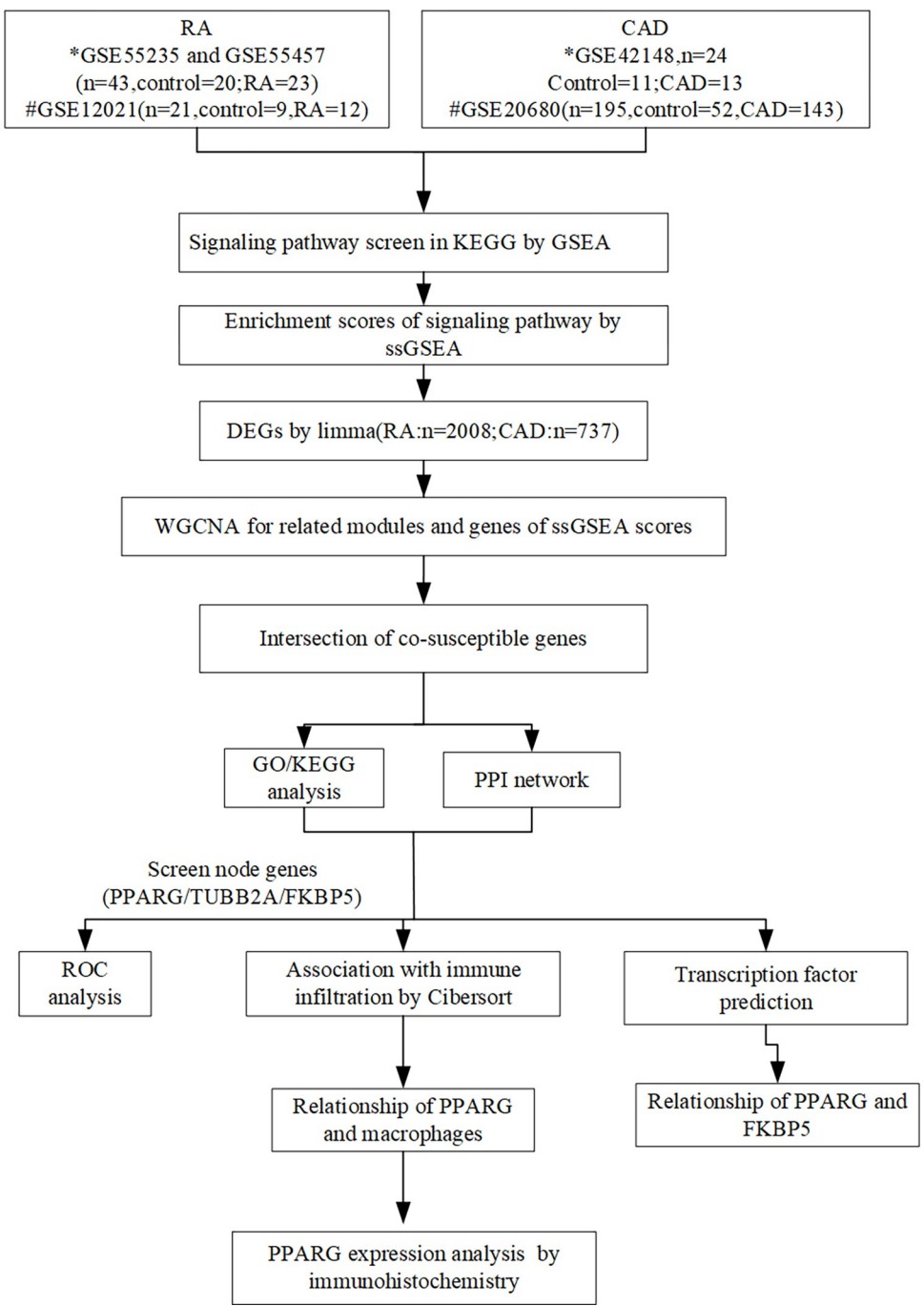

**Fig 1. Data analysis flowchart.** Schematic flowchart of data acquirement, processing, analysis, and validation.

datasets for WGCNA analysis using the R package (v1.71) [14]. Modules with a correlation coefficient > 0.7 were combined into one module.

## PCA and SVM analysis

To validate the discriminatory ability of 25 core genes in distinguishing a disease group from a normal group, the expression matrix was extracted from the training and validation datasets of RA and

CAD. PCA analysis was performed on the new datasets using R packages FactoMineR and factoextra. SVM analysis was then conducted using functions from the R package e1071. Finally, the classification performance of the 25 core genes was visualized by ROC curves using the pROC package.

## GO/KEGG analysis

Subsequently, gene ontology (GO) and KEGG enrichment were performed for co-susceptibility genes in both diseases by Clusterprofiler (v4.4.4) [15]. The STRING database (v11.0) [16] was used to construct a protein interaction network for RA and CAD co-susceptible genes, and the Cytoscape (v3.9.2) [17] plugin Molecular Complex Detection (MCODE) was applied to analyze clustering modules in the PPI network. Then, the four topology analysis algorithms MCC, MNC, Degree, and EPC in the cytoHubba plug-in were used to predict and explore the first five important hub genes in the PPI network. The candidate hub genes were obtained from the intersection genes of the four algorithms. The Disease Validation Sets (CAD: GSE20680; RA:GSE12021) were used to verify the differential expression of the above candidate hub genes by a Wilcoxon rank sum test between the disease and normal group.

## qRT-PCR

Three synovial tissue samples from RA patients who underwent joint replacement surgery and three artery specimens from patients with coronary artery disease undergoing vascular surgery were collected in 2022, which was approved by the Ethics Committee of the First Affiliated Hospital of Fujian Medical University (no. [2015]084–2). Total RNA was isolated from frozen sections of six matched pairs with RA and CAD lesion and adjacent normal tissue using TRIzol (Invitrogen, Carlsbad, CA, USA), and converted to cDNA with the PrimeScript RT-PCR Kit (TakaraBio, Otsu, Japan). The qRT-PCR was conducted using SYBR Select Master Mix on an ABI 7500 real-time PCR system (Applied Biosystems, CA, USA). The primer sequences were designed as follows. GAPDH was used as reference gene.Gene expression was calculated according to the 2-ΔΔCT method.

| gene | Forward | Reward |
|------|---------|--------|
| TUBB2A | CGCAGCCGGCACCAT | TGACCTCCCAAAACTTGGCG |
| FKBP5 | GCGTCCCAGAGGGGGAA | CTGGGGATTGTCGCTTCGTA |
| PPARG | AGAGCCTTCCAACTCCCTCA | TCTCCGGAAGAAACCCTTGC |
| GAPDH | GAAAGCCTGCCGGTGACTAA | CTGGGGATTGTCGCTTCGTA |

## Evaluation of the diagnostic efficacy of the Hub gene

Based on the gene expression data in the datasets, the ROC curve was plotted with R package pROC (v1.18.0) [18] to assess the diagnostic accuracy of the hub genes. The higher the AUC value, the stronger the diagnostic value in both diseases. Then the GeneMANIA online database (http://genemania.org/) was used to analyze the 20 interacting genes of the three hub genes. And the ChEA3 platform was used to explore common transcription factors (TFs) of key genes and the target gene-TF regulatory network was further visualized by Cytoscape software [19].

## Immune infiltration analysis of key genes

The immune cell infiltration of all datasets of both diseases was estimated with the CIBERSORT algorithm; the relationship between the hub gene and 22 immune cells was analyzed with a Pearson correlation and visualized via a lollipop plot.

### Immunohistochemical analysis of PPARG in synovium of RA and vessel of CAD

The design of this study was approved by the Ethics Committee of the First Affiliated Hospital of Fujian Medical University. Informed consent was exempted from all participants ([2015] 084–2).Surgical specimens of synovial tissue (n = 30) and atherosclerotic vascular wall (n = 30) from patients with RA and CAD were obtained from archived paraffin tissue in the pathological department of the First Affiliated Hospital of Fujian Medical University, covering the period from January 1, 2016, to December 31, 2022. Adjacent normal tissue served as control. The immunohistochemical staining for anti-CD68 (Dako, Clone PGM1, dilution 1:200) and anti-PPARG (Origene, Clone: TA312601, dilution:1:200) using a standard protocol was performed on an automatic immunohistochemical machine (Leica, Bond max), PBS served as negative control. PPARG expression was evaluated by its localization and staining area in different tissues. The percentage of staining area was calculated by Image J software.

## 3 Results

GSEA analysis identified two common activated signaling pathways in CAD and RA

To identify the common inflammation signaling pathways activated in both diseases, GSEA enrichment analysis was performed on the training datasets of CAD and RA diseases respectively. There are 16 pathways with a significant enrichment in CAD (Fig 2A), and 20 pathways with significant enrichment in RA (Fig 2B). Fc_gamma R mediated phagocytosis and allograft rejection is indicated within the intersection of the two pathways. (Fig 2C). The enrichment scores of the two intersection pathways in CAD and RA were calculated based on the ssGSEA (or GSVA) algorithm. Differential ssGSEA scores of the two signaling pathways in CAD and RA are shown with boxplots in Fig 2D.

### Identification of overlapped DEGs in CAD and RA

Using the R package limma (v3.10.3), 737 differential expression genes (DEGs) were identified, with 307 downregulated and 430 upregulated genes between normal and diseased groups in CAD datasets (Fig 3A). In addition, 2008 DEGs with 949 downregulated and 1059 upregulated genes were found in RA datasets (Fig 3B). Differential expression of 25 hub genes in the training and validation datasets were identified following WGCNA analysis, validated by a Wilcoxon test, and visualized with bean plots that are grouped by normal and disease as shown in Fig 3C–3F.

WGCNA analysis for two common pathways-related module genes in CAD and RA

As for CAD, differential genes identified in GSE42148 set were used in constructing the WGCNA network. To identify key genes and co-expression modules associated with the Fc_gamma receptor mediated phagocytosis pathway, a scale-free topology model fit was employed to determine an appropriate soft threshold at power of 10 with $R^2 = 0.9$ (Fig 4A). Through the hierarchical clustering dendrogram, six distinct co-expression modules were identified except the grey module, which cannot be classified into any module (< 5 genes), as shown in Fig 4B. The important modules were identified according to a high correlation between these co-expression gene modules and sample traits ($|r| \geq 0.7$, $p < 0.05$), which are blue and black modules (Fig 4C). There are total 296 DEGs for WGCNA analysis.

With regard to RA, soft-thresholding powers were defined using the pickSoftThreshold function of the WGCNA R package, and beta (β) = 7 (scale-free $R^2 = 0.9$) was selected to be appropriate for soft threshold power (Fig 4D). A total of 2008 DEGs were analyzed and six co-expressed gene modules identified, the blue, magenta, black, and salmon modules were most

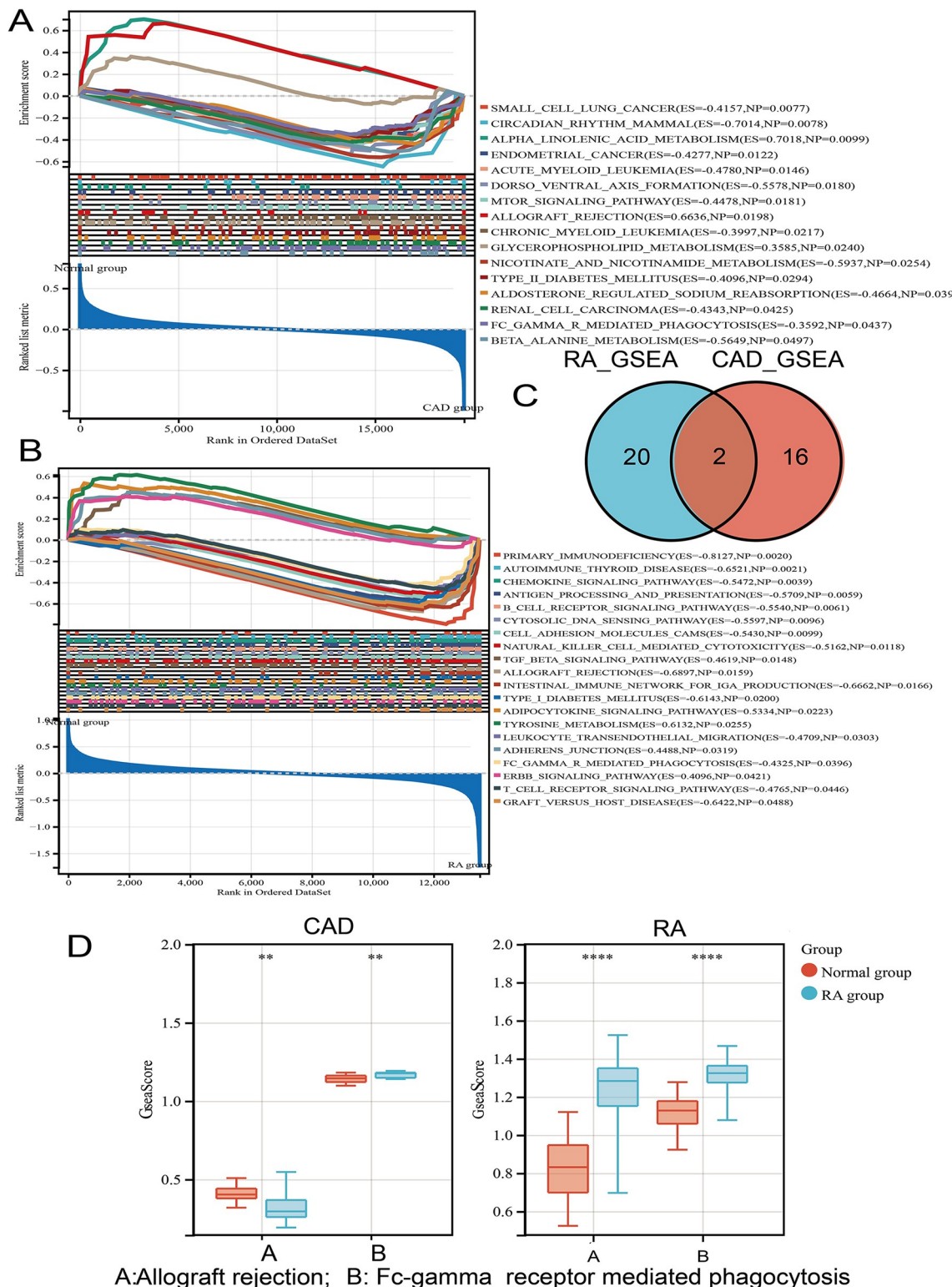

**Fig 2. GSEA analysis in the RA and CAD training datasets. A-B** Multi-GSEA plot showing 16 significantly differential pathways for CAD and 20 RA significantly differential pathways; **C** Venn diagram showing intersection of CAD and RA related pathways; **D** ssGSEA score of allograft rejection and Fc gamma receptor mediated phagocytosis displayed in boxplots.

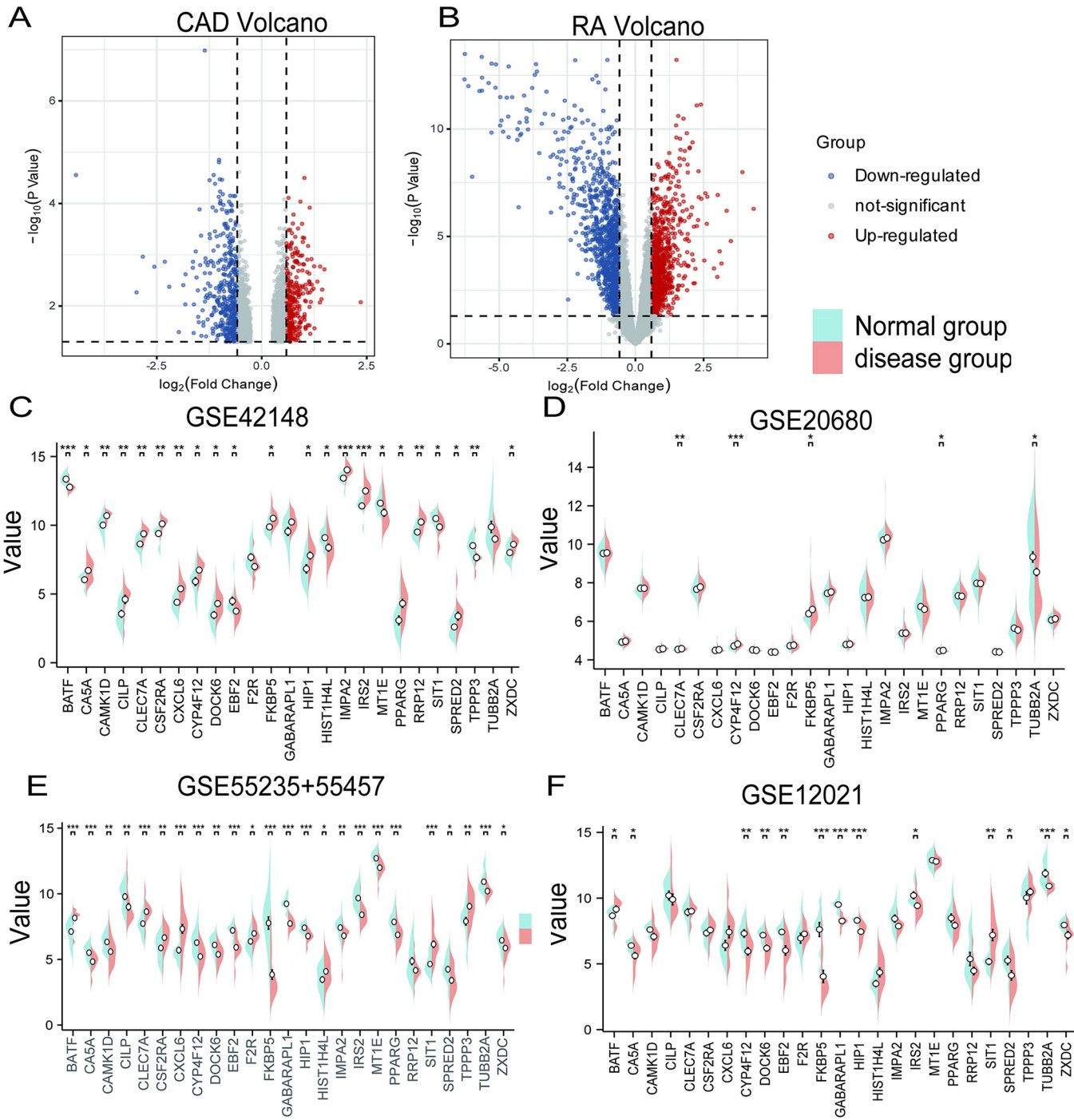

**Fig 3. Differential expression analysis of CAD and RA datasets. A-B** Volcano plots of DEGs in RA and CAD training datasets; **C-F** Bean plots of the 25 common DEGs in CAD and RA training and validation datasets.

tightly correlated to the Fc _gamma receptor mediated phagacytosis pathway ($|r| \geq 0.7$, $p < 0.05$) (Fig 4E and 4F). Based on the intersection of WGCNA module genes obtained for theRA(1585) and CAD (271) diseasesmentioned above, 25 co-susceptible genes were identified, as seen in the Venn diagram and bean plots in Fig 2G.

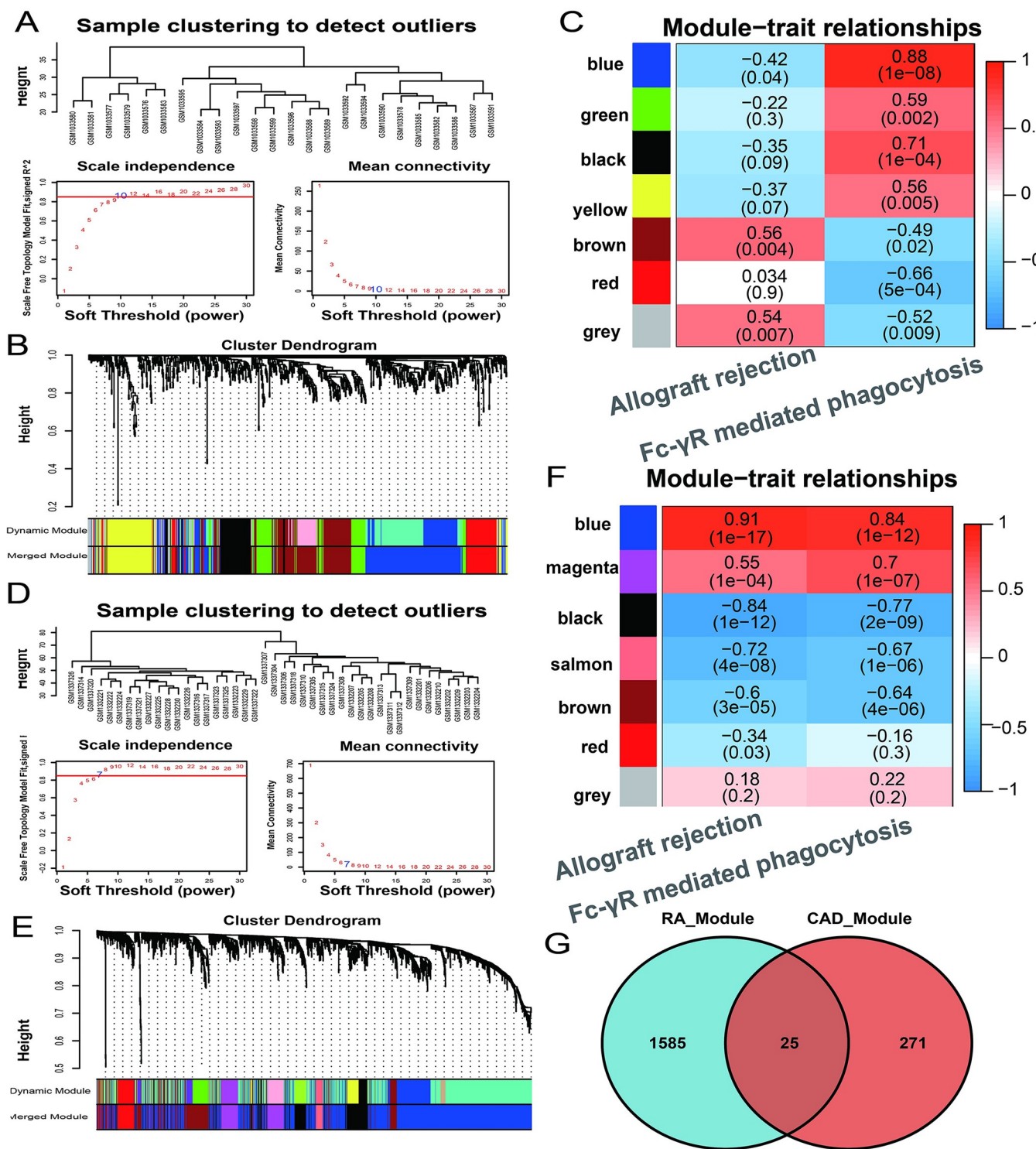

**Fig 4. WGCNA analysis for CAD and RA training datasets. A and D** A soft threshold determination plot was generated for CAD and RA training datasets, with an optimal power of 10 for CAD and 7 for RA. **B and E** WGCNA module identification and clustering dendrogram of DEGs in CAD and RA training datasets. **C and F** The module -trait correlation heatmap in WGCNA of CAD and RA training datasets. Each row represents a gene module, while each column represents ssGSEA score of a significantly differential pathway. The number within the heatmap indicates the correlation coefficient and *p* values. **G** Venn plot shows the intersection of module genes of WGCNA in CAD and RA datasets.

### The classification and prediction results of 25 core genes by PCA and SVM analysis

PCA analysis revealed that the 25 core genes effectively distinguished the disease group from the normal group in both the RA and CAD training and validation datasets (Fig 5A–5D). SVM analysis showed threshold values of 0.568 and 0.608 for the RA training and validation datasets, with AUCs of 1.00 and 0.991 respectively. For the CAD training and validation datasets, the threshold values were 0.678 and 0.937, with AUCs of 0.993 and 0.843, respectively (Fig 5E–5H). These results indicate the 25 core genes hold good classification potential for both the disease and normal groups in RA and CAD.

### GO and KEGG pathway analysis of 25 susceptible genes in RA and CAD

The 25 susceptible genes obtained in the previous steps were subjected to GO-Biological Process (GO-BP), GO-Cellular Component (GO-CC), GO-Molecular Function (GO-MF), and KEGG pathway enrichment analysis using the R package ClusterProfiler. A total of 19 GO-BP, 9 GO-CC, 45 GO-MF, and 6 KEGG pathway enrichment were identified.The top 9 GO enrichment and KEGG pathways based on $p$ value were selected, and displayed in Fig 6A–6D.

### PPI network construction of 25 susceptibility genes and hub genes identification

Using the STRING database, a protein interaction network for the 25susceptible genes of CAD and RA was constructed (Fig 6E). Cytoscape plugin MCODE for module analysis was employed to detect key clustering modules in the PPI network. Then, four topological analysis algorithms, MCC, Degree, EPC, and BottleNeck, from the cytoHubba plugin, were used to explore the top five important hub genes in the PPI network. The intersection of the five genes obtained by the four algorithms includes three genes, PPARG, FKBP5, and TUBB2A, as shown in Fig 6F. qRT-PCR showed that PPARG had low expression in RA tissues ($p < 0.05$) and high expression in CAD tissues compared to their respective normal tissues ($p < 0.001$). TUBB2A was lower in RA tissue ($p < 0.01$) but showed no differential expression in CAD tissues, while FKBP5 was higher in CAD tissue ($p < 0.001$) but did not show differential expression in RA tissues (Fig 6G and 6H).

### ROC evaluation of the predictive ability of the three hub genes in RA and CAD

The diagnostic potential of three biomarkers, TUBB2A, FKBP5, and PPARG was assesed for risk of both diseases. ROC analysis was conducted across all datasets, revealing AUC values of over 0.6 for the three hub genes across the training and validation sets. PPARG, TUBB2A, and FKBP5 demonstrated comparable diagnostic efficacy for both RA and CAD (Fig 7A-7D), indicating the potential significance of these markers for RA and CAD prediction, although further studies are required to authenticate these results.

### PPI network construction for three hub genes

To investigate the functional association of the three candidate genes, the Gene Multiple Association Network Integration Algorithm analysis (GeneMANIA) was performed online. The core genes were used as query genes on GeneMANIA to produce a network between them. Most of the network interactions were physical interactions, genetic interactions or co-expressions. The largest functional group genes in the network were related to ligand activated

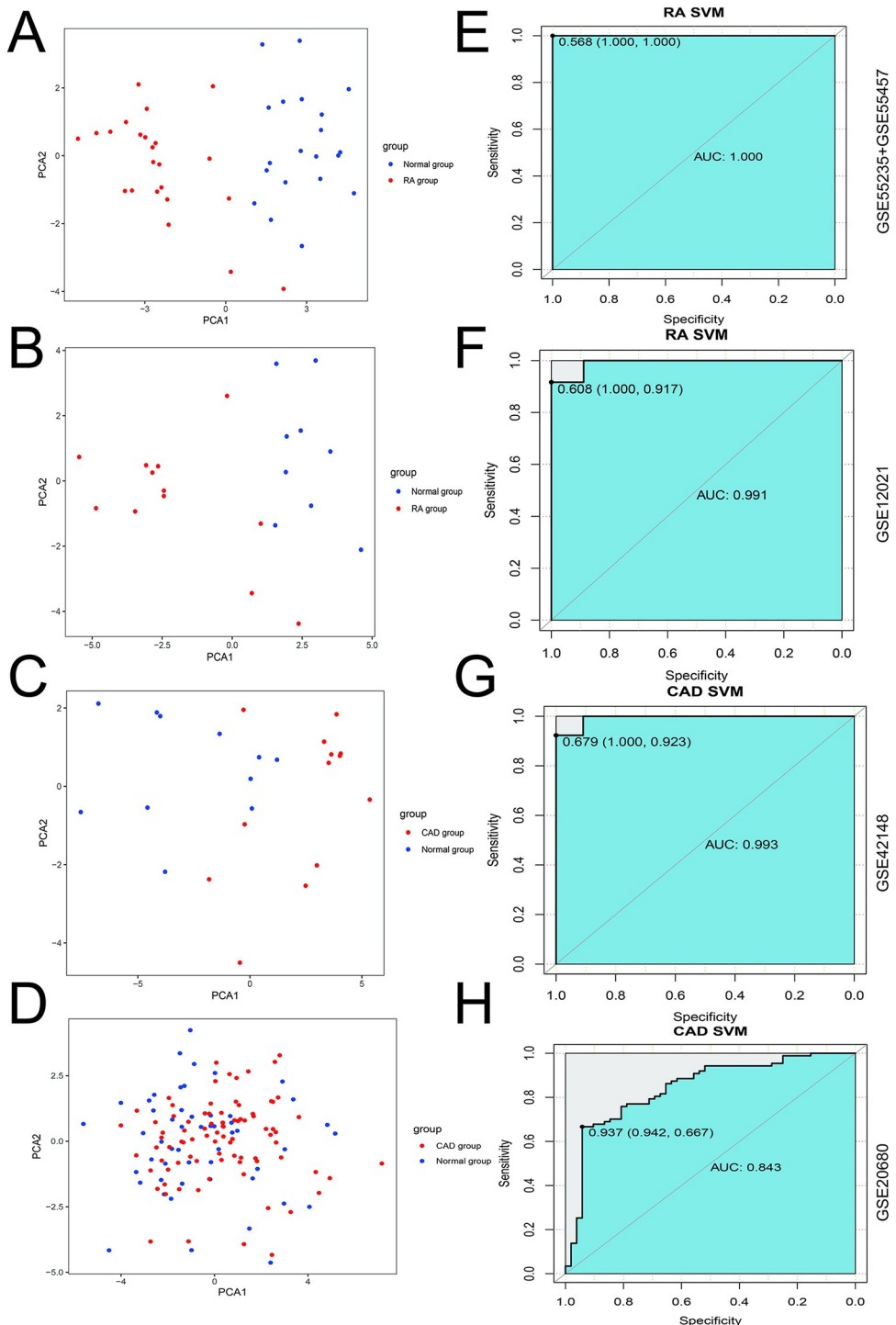

**Fig 5. PCA and SVM analysis for 25 core genes in RA and CAD datasets. A-D** PCA scatter plot between normal and disease groups for training and validation datasets of RA and CAD. **E-H** ROC curves and their corresponding AUCs for SVM methods.

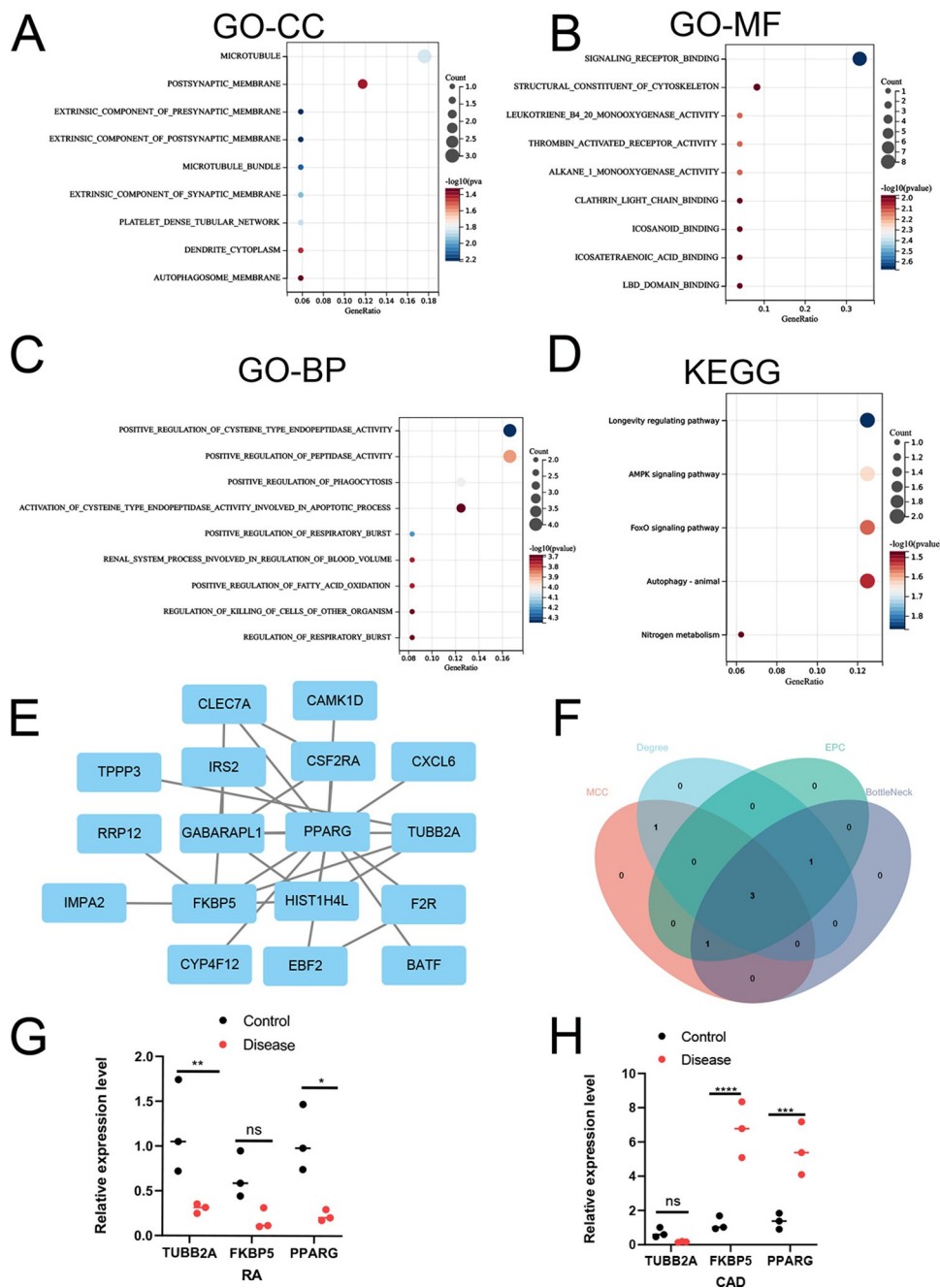

**Fig 6. Biological function analysis, PPI network and hub genes of common DEGs in CAD and RA training datasets. A-D** GO analysis and KEGG pathway enrichment analysis of the 25 DEGs. **E** PPI network construction of the 17 DEGs. **F** Venn plot showing the intersection of the three hub genes identified by four plug-ins of cytoscape. **G-H** qRT-PCR analysis of mRNA expression of TUBB2A, FKBP5, and PPARG in RA and CAD tissues with their matched normal tissues as control. Each sample was repeated three times and the results are expressed as mean ± SD. (ns, non-significant, $^{*}p < 0.05$, $^{**}p < 0.01$, $^{***}p < 0.001$. Student's t-test).

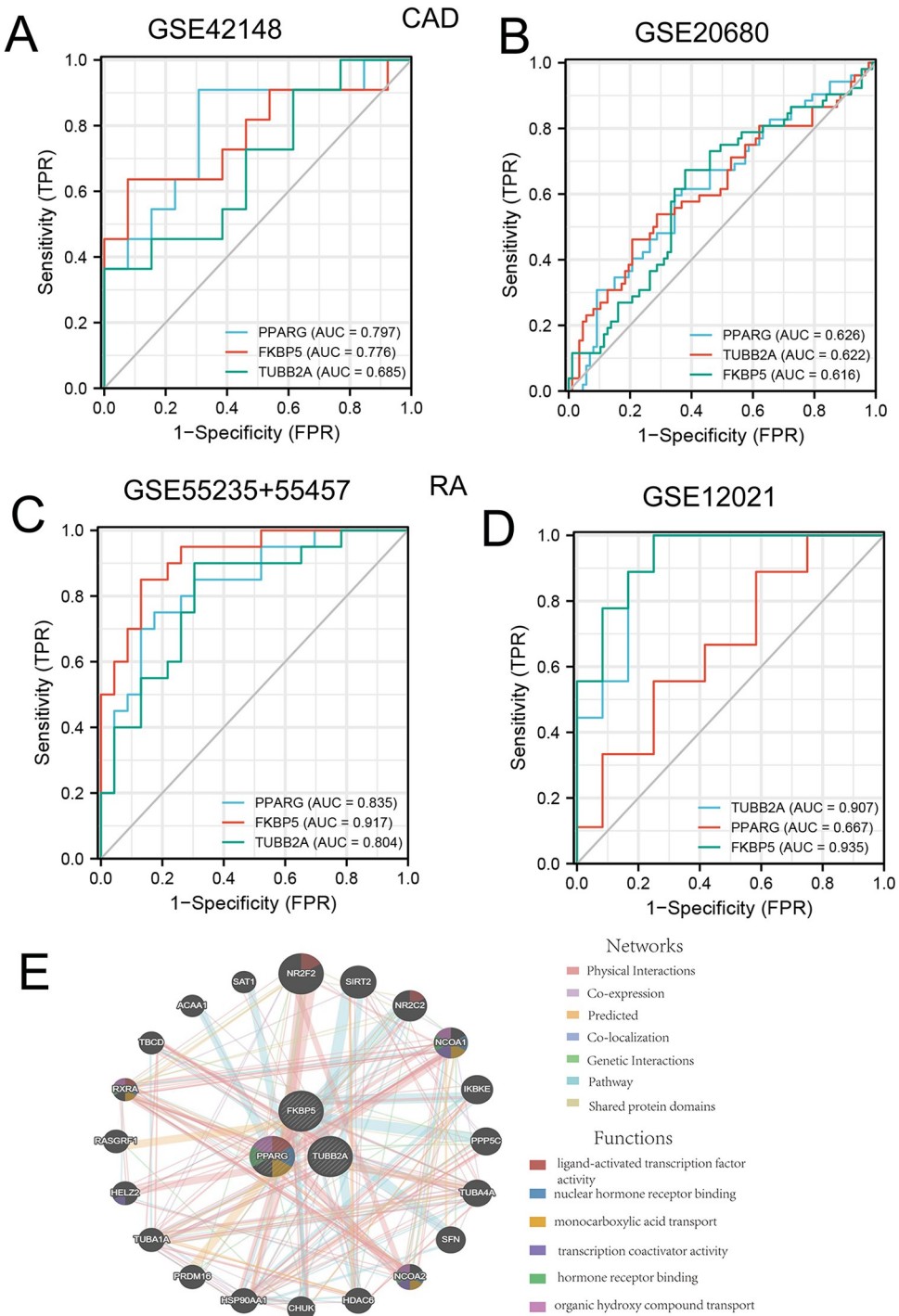

**Fig 7. Assessment of diagnostic efficacy of three hub genes. A-B** The ROC curves of the training and validation cohorts of CAD; **C-D** The ROC curves of the training and validation cohorts of RA; **E** Gene-gene interaction network construction for TUBBB2A, FKBP5, and PPARG by GeneMANIA.

transcription factor activity. The involved genes included PPARG, NR2F2, NR2C2, and RXRA, indicated by red color (Fig 7E).

## Correlation analysis between hub genes and immune cell infiltration

The immune cells of all training datasets of RA and CAD were calculated based on the CIBER-SORT algorithm. As shown in Fig 8A–8D, the relationship between hub genes and 22 immune cells was analyzed using pearson correlation analysis, the gene PPARG was positively correlated with monocyte in CAD across the training and validation datasets (R = 0.742, p < 0.01 and R = 0.219, p < 0.05, respectively), which revealed a potential relationship of PPARG and monocyte in CAD. In the CAD datasets GSE20680, we also noticed that PPARG in peripheral blood was inversely correlated with M2 macrophages (R = -0.247, p < 0.05). On one hand, PPARG expression in synovial epithelium positively correlated with CD4 memory resting T

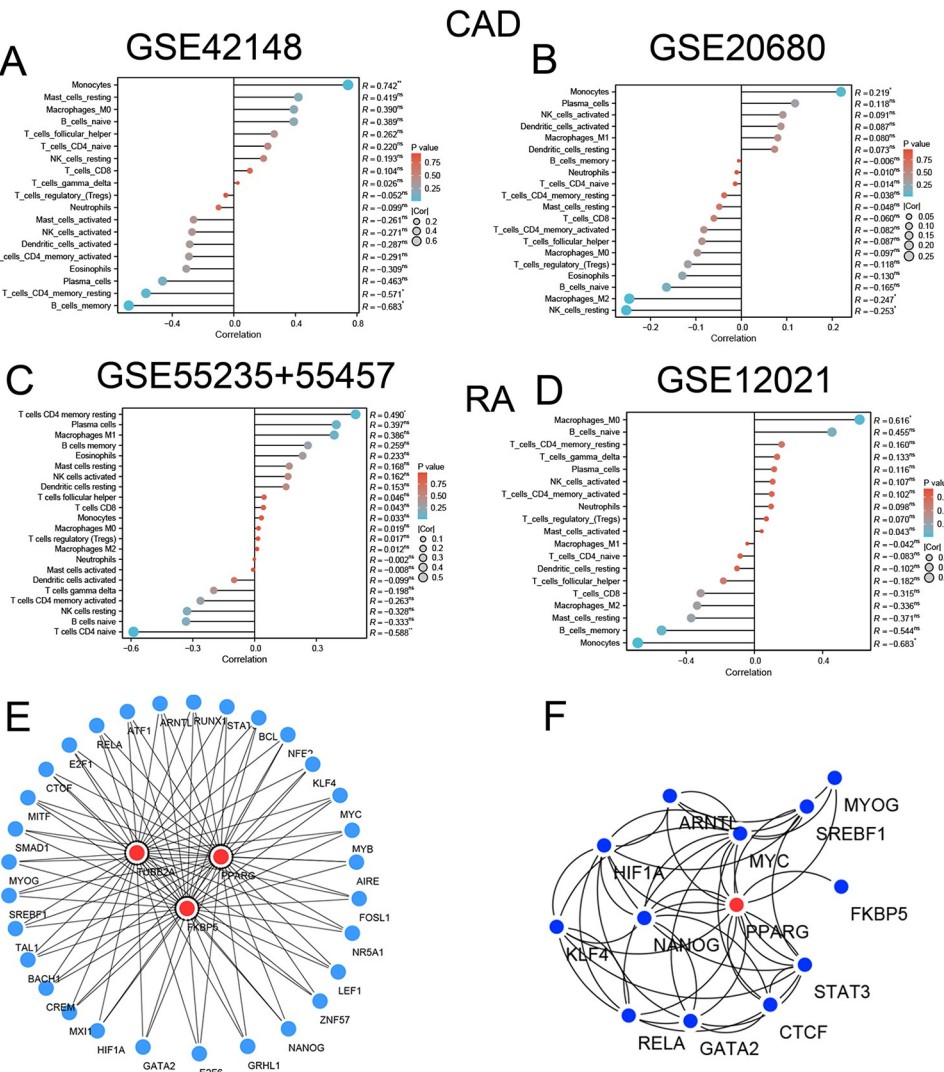

**Fig 8. Correlation analysis, immune cell infiltration, and transcription network construction of PPARG. A-D** Lollipop diagrams showing the relationship of PPARG expression and 22 immune infiltration scores in RA and CAD datasets. **E** Transcription factor enrichment analysis by ChEA3 database and PPI network construction for TUBB2A, PPARG, and FKBP5. **F** Cytoscape analysis of first-order interactions involving PPARG relevant transcription factors.

cells (R = 0.490, p < 0.05) and negatively correlated with CD4 naive T cells (R = -0.588, p < 0.01) in RA training datasets. On the other hand,a positive correlation was identified between PPARG and M0 macrophages (R = 0.616, p < 0.05), while a negative correlation was observed between PPARG and monocyte (R = -0.683, p < 0.05). Thus we infer that PPARG may be involved in the regulation of macrophages in both diseases, although the exact mechanisms underlying these associations need further research.

### Shared transcription factor network of three hub genes

The ChEA3 database was used to predict transcription factors for the three hub genes. A total of 30 transcription factors associated with the hub genes were identified. The TF-target network was constructed using Cytoscape software, as shown in Fig 8E, and PPARG first-order correlated transcription factors were listed separately in Fig 8F, which shows an interactive relationship of PPARG and FKBP5 in RA linked CAD. Moreover, spearman correlation analysis demonstrated positive correlation of PPARG and FKBP4 in the GSE42148 datasets of CAD (R = 0.570, p = 0.004, Fig 8L).

### PPARG expression down-regulated in synovial epithelium of RA and up-regulated in foamy macrophages of CAD

The histological expression of PPARG were examined by immunohistochemistry. In contrast with the control group, strong staining of PPARG was found in foamy macrophages as indicated by CD68 positivity in the artery wall of CAD with atherosclerotic plaque formation being observed (Fig 9A–9F). The Pearson correlation coefficient of both CD68 and PPARG was 0.6276 (p = 0.002) (Fig 9G). PPARG showed mild expression in control synovial epithelium compared with the RA patients, which showed little expression of PPARG, especially in the foci of aggregated lymphocytes and plasma cells in the sub-synovial epithelium (Fig 9H–9K). Pearson correlation analysis also demonstrated positive relationship of PPARG and FKBPF in CAD datasets GSE42148(Fig 9L), which may hinder their potential interaction in the development of CAD.

## 4 Discussion

This study focused on the shared molecular pathways and key genes involved both in RA and CAD.Using WGCNA and Cytohubba methods three hub genes (TUBB2A,FKBP5 and PPARG)were identified that are commonly activated in both diseases. These genes were further analyzed using ROC analysis, demonstrating their potential diagnostic value for RA and CAD.Finally, immune infiltration patterns were analyzed by CIBERSORT algorithm, which revealed the correlation between PPARG and mononuclear cells or macrophages, highlighting its roles in the immune mechanisms of both diseases. Co-expression and TF-mRNA regulatory network construction further provide new insights to the potential biological roles of PPARG in the common pathogenesis of both diseases.

RA and CAD are both complex inflammatory diseases. Macrophages, as key immune cells, play a significant role in the pathogenesis of both conditions. On one hand, macrophages fuel synovial inflammation through the release of pro-inflammatory cytokines like tumor necrosis factor-alpha (TNF-a) and interleukin-1(IL-1) [20]. Their infiltration in the synovial membrane pannus can cause cartilage and bone destruction by releasing angiogenic factors and excreting matrix metalloproteinases (MMPs) [21]. On the other hand, macrophages internalize lipids and transform to foamy cells to initiate the formation of atherosclerotic plaque in arterial walls [22]. These macrophages also contribute to plaque rupture and subsequent thrombosis by releasing MMPs and activating clotting factors [23]. Current research indicates the phagocytic

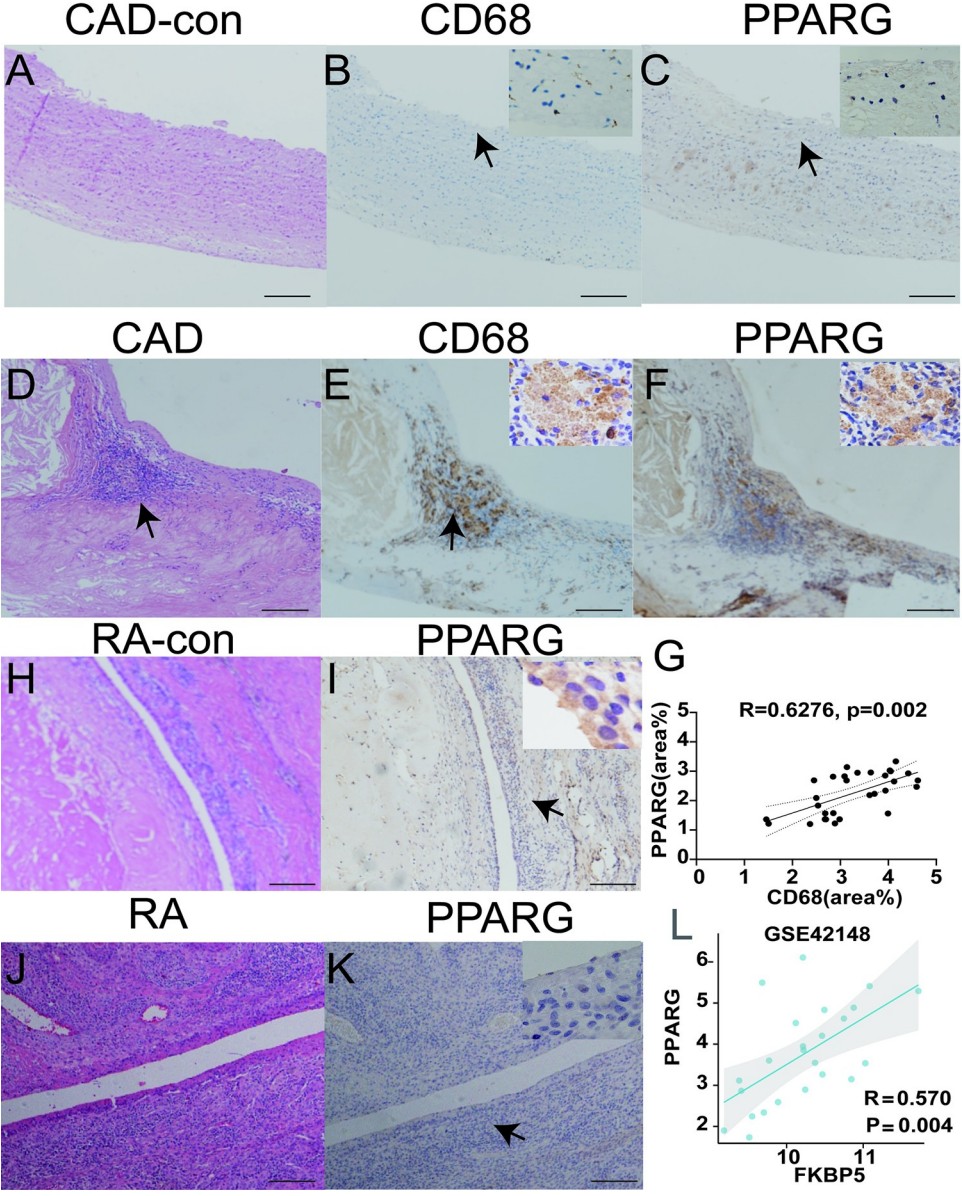

**Fig 9. Immunohistochemistry analysis of PPARG and CD68 expression on tissue samples from patients of CAD/RA. A-C** In the control group, there is little expression of CD68 and PPARG on the arterial wall; **D-F** Both CD68 and PPARG were observed to be co-expressed in the foamy histiocytes of CAD patients with strong brown-yellow cytoplasmic staining. H.E. staining showed cholesterol crystals observed in the upper left corner; **G** The percentage of positive areas for CD68 and PPARG were analyzed by Image J and presented with simple linear regression. **H-I** PPARG exhibited mild staining intensity in control synovial epithelium and little expression in the sub-synovial interstitial cells. **J-K** The synovial epithelium of RA patients exhibited significant infiltration of lymphocytes and plasma cells in the sub-synovial region with little expression of PPARG. Bar = 50 um, IHC and H.E. magnification, 400×; **L** Positive relationship of PPARG and FKBPF in CAD datasets GSE42148.

activity mediated by the Fc_gamma receptor is an intersected pathway for both RA and CAD. Fc_ gamma receptors are integral to macrophage function and aid in the regulation of macrophage phagocytosis [23]. The clearance of pathogens and cellular debris helps to maintain immune homeostasis and prevent infections. However, Fc_gamma receptors also activate macrophages to promote the release of inflammatory factors and enhance immune response

[24]. Thus, Fc_ gamma receptors are a double-edged sword in macrophage function and the regulation of macrophage phagocytosis.

Significant correlative modules with Fc-gamma receptor-mediated phagocytosis phenotype were identified in both RA and CAD training datasets using a WGCNA analysis. Biological function analysis indicated the 25 DEGs within these modules in RA and CAD datasets may show discriminative ability for normal and disease groups as well as links to several immune related signaling pathways such as ligand-receptor binding [25], endocytosis [23], microtubule [26] and classical pathways like AMPK [27] and FOXO [28]. These findings suggest these genes play important roles in regulating immune responses and may have significant implications in the inflammatory pathogenesis of RA and CAD. The Cytohubba plug-in revealed three core genes (TUBB2A, PPARG, and FKBP5) among the 25 DEGs. Tubulin beta class IIa (TUBB2A) is encoded in the 10q24.2 locus, which plays a crucial role in cell division, cell motility, intracellular transport, and immunomodulatory processes [29]. Increased TUBB2A expression has been reported in intestinal cells during inflammation-associated bowel disease, thus highlighting its potential as a therapeutic target in immune modulation [30]. Peroxisome proliferator-activated receptor gamma (PPARG) is a transcription factor located on 3p25.2 [31] that is involved in regulating macrophage polarization, lipid metabolism and inflammation [32]. FK506-binding protein 5 (FKBP5) is situated on 6p21.31[33], which binds to glucocorticoid receptors, playing a critical role in regulating cellular stress response, metabolism, and immune response [34]. qRT-PCR results showed that lower and higher mRNA expression of PPARG in RA and CAD tissues compared with their normal tissues, which is consistent with DEGs results from bioinformatics analysis. However, TUBB2A and FKBP5 mRNA expression may need more samples for further validation. Recent studies have shown the significance of the three genes in the pathogenesis of RA and CAD. Li XF et.al. demonstrated PPARG importance for proliferation and migration of fibroblast-like synoviocytes in RA [35]. Likewise, PPARG SNP rs3856806 has been reported to be associated with increased risk for atherosclerotic disease in the Asian population [36]. TUBB2A has been reported to be overexpressed in peripheral blood of postpartum onset RA [37]. However, there is little literature about correlation of TUBB2A and CAD. The FKBP5 SNP locus is linked to increased CAD risk [38]. Based on the ROC analysis, PPARG, TUBB2A, and FKBP5 demonstrate comparable efficacy in discriminating between the normal and disease groups for both RA and CAD. In the GSE12021 dataset of the RA validation cohort, PPARG showed a relatively lower AUC. Nonetheless, it was decided to further investigate the relationship between PPARG and immune infiltration due to its strong association with the mononuclear/macrophage system, as indicated by reference [39] and our own analysis using the CIBERSORT algorithm.

PPARG, a gene associated with lipid metabolism, showed upregulation, and positively correlated with peripheral blood mononuclear cells (PBMCs) in CAD. The association between PPARG and CAD remain controversial from previous studies. Initial investigations suggest that PPARG promotes the phagocytosis of lipids by monocyte, leading to the formation of foamy cells and lipid plaques [40]. However, recent studies indicate that PPARG enhances insulin sensitivity, promotes adipogenesis, and exerts anti-inflammatory and anti-atherosclerotic effects [41]. However, there was no statistically significant correlation between PPARG expression and M0/M1 macrophages. Further immunohistochemical staining showed little PPARG expression in relatively normal vessel walls and moderate PPARG expression in CD68-positive foamy macrophages of the vascular walls of CAD. These findings are consistent with the finding by Tontonoz. P [40]. Moreover, PPARG expression negatively correlated with M2 macrophages in the validation set, which are known to have anti-inflammatory effects. This suggests that PPARG may have a positive role in regulating the lipid induced inflammation in CAD.

However, the role of PPARG in PBMC activation is complex and requires further investigation.

In the RA validation set, a positive relationship was observed between PPARG and M0 macrophages; however, in the RA training set, an association was discovered with memory resting CD4+T cells. The immunohistochemistry analysis illuminates the cytoplasmic expression of the macrophages of control healthy groups (type A synovial cells). However, it is scarcely expressed in the synovial epithelial of RA, particularly in the area with aggregation of lymphocytes and plasma cells beneath the synovium. Given that both M0 and memory resting CD4 + T cells are pre-activated cells, we postulate that PPARG expression in the normal synovial epithelium may contribute to sustain the dormant state and immune quiescence of the two cell types [42]. Its role in the development of RA appears to be negligible.

Our preliminary investigation highlighted FKBP5 as a key candidate gene for RA and CAD. The TF-mRNA network and significantly positive correlation of PPARG and FKBP5 in GSE42148 dataset of CAD suggest a potential interaction with PPARG within the regulatory network. Notably, FKBP5 has been identified as a central component in controlling the natural restorative regulation of mitophagy via PPARG in pathological demyelinating settings [43]. This begs the need for future research aimed to empirically validate the relationship between FKBP5 and PPARG in the development of CAD.

This study aims to identify and assess the diagnostic efficacy of specific genes in identifying RA complicated CAD. However, it is important to consider certain limitations that may impact the interpretation of the results. The use of peripheral blood samples for CAD and synovial tissue samples for RA introduces inherent differences in their biological properties and gene expression profiles. In the retrospective study, collecting adequate numbers of blood samples from patients with RA complicated CAD to assess the expression levels of PPARG posed challenges. However, immunohistochemical staining of synovial and arterial wall samples obtained from patients with RA and CAD, respectively, demonstrated a specific correlation between PPARG expression and type A synovial epithelium, as well as foamy histiocytes with macrophage lineage. Despite these limitations, this study provides valuable insights into the potential diagnostic efficacy of PPARG in RA and CAD which showed down-regulated expression in synovial tissues of RA and up-regulated expression in PBMCs. Future studies with larger sample sizes are warranted to further validate and expand upon these findings.

## 5 Conclusion

In conclusion, activation of the Fc_gamma receptor mediated signaling pathway is a shared characteristic of RA and CAD. WGCNA analysis identified three hub genes (TUBB2A, FKBP5, and PPARG) displaying effective predictability in distinguishing normal from disease group for both RA and CAD. Further analysis revealed that PPARG is associated with PBMCs involved in CAD development. This finding provides new insights into the common pathogenesis of RA and CAD,with PPARG emerging as a potential predictive marker for RA linked CAD.

## Supporting information

**S1 Raw data.**
(DOCX)

## Acknowledgments

We thank Shenglin Lin from Departemnt of Biomedical Informatics of Fujian Medical University and Dr. Fahui Liu from Cell Therapy Research Center of the First Affiliated Hospital of Xiamen University for valuable suggestion of bioinformatics analysis.

## Author Contributions

**Conceptualization:** Zhenzhen Zhang, Sheng Zhang, Xia Zhu.

**Data curation:** Zhenzhen Zhang.

**Formal analysis:** Zhenzhen Zhang.

**Funding acquisition:** Zhenzhen Zhang, Xia Zhu.

**Investigation:** Zhenzhen Zhang.

**Methodology:** Yupeng Chen, Xiaodan Fu, Junlan Wang, Qingqiang Zheng.

**Visualization:** Zhenzhen Zhang.

**Writing – original draft:** Zhenzhen Zhang.

**Writing – review & editing:** Linying Chen, Sheng Zhang, Xia Zhu.

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
