## [Decision Letter · Decision Letter 0]

28 Sep 2023

PONE-D-23-28394Identification of PPARG as key gene to gap CAD and RA via microarray data analysisPLOS ONE

Dear Dr. zhang,

Thank you for submitting your manuscript to PLOS ONE. After careful consideration, we feel that it has merit but does not fully meet PLOS ONE’s publication criteria as it currently stands. Therefore, we invite you to submit a revised version of the manuscript that addresses the points raised during the review process.

We look forward to receiving your revised manuscript.

Kind regards,

Abozar Ghorbani, Ph.D

Academic Editor

PLOS ONE

7. We note that Figure 8A, 8B, 8C, 8D, 8E, 8F, 8H, 8I, 8J and 8K in your submission contain copyrighted images. All PLOS content is published under the Creative Commons Attribution License (CC BY 4.0), which means that the manuscript, images, and Supporting Information files will be freely available online, and any third party is permitted to access, download, copy, distribute, and use these materials in any way, even commercially, with proper attribution. For more information, see our copyright guidelines: http://journals.plos.org/plosone/s/licenses-and-copyright.

A. You may seek permission from the original copyright holder of Figure 8A, 8B, 8C, 8D, 8E, 8F, 8H, 8I, 8J and 8K to publish the content specifically under the CC BY 4.0 license. 

B. If you are unable to obtain permission from the original copyright holder to publish these figures under the CC BY 4.0 license or if the copyright holder’s requirements are incompatible with the CC BY 4.0 license, please either i) remove the figure or ii) supply a replacement figure that complies with the CC BY 4.0 license. Please check copyright information on all replacement figures and update the figure caption with source information. If applicable, please specify in the figure caption text when a figure is similar but not identical to the original image and is therefore for illustrative purposes only.

Additional Editor Comments:

1. Fig 1. Need more explanation in legend.

2. The text need to be improved as English Grammar and scientific frame.

3. Space and “,” should be checked in the whole text.

4. Some DE genes should be validated using qRT-PCR or other techniques. 

5. Quality of figures should be improved and the text and details should be clear.

Reviewers' comments:

Reviewer's Responses to Questions

**Comments to the Author**

1. Is the manuscript technically sound, and do the data support the conclusions?

Reviewer #1: Yes

Reviewer #2: Yes

2. Has the statistical analysis been performed appropriately and rigorously? 

Reviewer #1: N/A

Reviewer #2: Yes

3. Have the authors made all data underlying the findings in their manuscript fully available?

Reviewer #1: Yes

Reviewer #2: Yes

4. Is the manuscript presented in an intelligible fashion and written in standard English?

Reviewer #1: Yes

Reviewer #2: Yes

5. Review Comments to the Author

Reviewer #1: This study identifies 737 DEGs associated with diseased groups in CAD datasets and 2008 DEGs associated with diseased groups in RA datasets. The author observed that TUBB2A, FKBP5, PPARG were significantly associated with RA and CAD. Major revisions are requested for this paper to be accepted.

Major comments:

1. Although the authors made some efforts attempting to integrate several existing gene expression and datasets via GO and GSEA enrichment that they chose for over-represented functional gene sets, validation experiments that could support the authors' conclusion are missing, sufficient new/validated information is lacking.

2. This study need more work in order to identify more datasets or use data from authors' laboratory to show replication of the results.

3. The authors need to use a PCA and a regression SVM (R-SVM) to obtain a global view and evaluate the performance of DEGs in classifying control conditions versus CAD and RA conditions.

4. Lines 75 to 85 are not related to the introduction.

5. Most of the Figures are not clear eg, Fig 2 (A,B), Fig 4 (A,D,E), Fig 5 (A,B,C,D,F) , Fig 6 (E) and Fig 7 (A,B,C,D,E). The authors provide clearer Figures.

Reviewer #2: It was a valuable study. However, some punctuation and grammatical errors are included within the manuscript. Therefore, with a minor revision and correction of the mistakes, this manuscript is absolutely suitable and recommended for publication.

6. PLOS authors have the option to publish the peer review history of their article (what does this mean?). If published, this will include your full peer review and any attached files.

Reviewer #1: No

Reviewer #2: No

---

## [Author Response · Author response to Decision Letter 0]

29 Dec 2023

Major comments:

1.Although the authors made some efforts attempting to integrate several existing gene expression and datasets via GO and GSEA enrichment that they chose for over-represented functional gene sets, validation experiments that could support the authors' conclusion are missing, sufficient new/validated information is lacking.

Dear Reviewer,

Thank you for your valuable feedback. 

In addition to bioinformatics analysis, our immunohistochemistry data validate the differential expression and localization of PPARG on synovial tissue and arterial wall samples from RA and CAD patients. The lack of validation in vivo and in vitro experiments in indeed the limitation of our study. Unfortunately, due to time constraints, reagent availability, and current financial limitations, we were unable to perform the validation experiments as part of this study.However, we want to assure you that the validation experiments are an integral part of our future research plan. 

Moreover, we speculated that the identified hub genes PPARG can gap RA and CAD through macrophages mediated Fc-gamma receptor-mediated endocytosis based on the following reference

 1 Chawla A, Barak Y, Nagy L, Liao D, Tontonoz P, Evans RM. PPAR-gamma dependent and independent effects on macrophage-gene expression in lipid metabolism and inflammation. Nat Med. 2001 Nov;7(1):48-52. doi: 10.1038/83336. PMID: 11135614. Which support PPARG function in lipid metabolism induced inflammation, including cornary artery disease.

2 Toobian D, Ghosh P, Katkar GD. Parsing the Role of PPARs in Macrophage Processes. Front Immunol. 2021 Dec 22;12:783780. doi: 10.3389/fimmu.2021.783780. PMID: 35003101; PMCID: PMC8727354. Which provide evidence to support that PPARG has been implicated in regulating macrophage polarization, phagocytosis, inflammatory cytokine production, and lipid metabolism .

3 Brosig L, Hong J, Wallis BB, Giacomini JC, Assimes TL, Goronzy JJ, Weyand CM. Hypermetabolic macrophages in rheumatoid arthritis and coronary artery disease due to glycogen synthase kinase 3b inactivation. Ann Rheum Dis. 2018 Jul;77(7):1053-1062. doi: 10.1136/annrheumdis-2017-212647. Epub 2018 Feb 3. PMID: 29431119. Which support that the macrophages implicated in the pathogenesis of both RA and CAD

2. This study need more work in order to identify more datasets or use data from authors' laboratory to show replication of the results.

Dear Reviewer, 

Thank you for your valuable feedback. 

In this study, we have carried out several verification steps. Firstly, 25 DEGs obtained from the intersection of DEGs in RA (GSE5523+GSE55457) and CAD (GSE42148) datasets and WGCNA were validated in validation datasets(RA GSE12021 and CAD GSE20680), referenced to (Fig.2)Additionally, we performed PCA and SVM analysis ( Fig.3 ) and ROC analysis to identify diagnostic discrimination ability in disease group and normal group of the 25 core genes and the three hub genes on both training and validation datasets(Fig.5) . Then qRT-PCR was conducted to validate the mRNA expression level of three hub genes in RA and CAD disease compared with their matched normal tissues. Finally, we conducted immunohistochemistry experiments to confirm the expression and localization of the core gene PPARG in synovial tissue and arterial wall macrophages from patients with chronic synovitis and coronary artery disease to support our bioinformatic study from clinical aspects (Fig.9)

We acknowledge the importance of replicating our findings to enhance the robustness of our study. However, we carefully considered various factors in our analysis, including sample size matching, sample types, result integration of the article logic, which led us to prioritize the mentioned datasets. Furthermore, the experimental validations will be supplemented in future studies and grant applications.

3.The authors need to use a PCA and a regression SVM (R-SVM) to obtain a global view and evaluate the performance of DEGs in classifying control conditions versus CAD and RA conditions.

Dear Reviewer, 

Thank you for your valuable feedback and suggestions.

We obtained the 25 core genes from the intersection of DEGs in RA (GSE5523+GSE55457) and CAD (GSE42148) datasets and WGCNA. To validate the discriminatory ability of 25 core genes in distinguishing disease group from normal group in both RA and CAD, the expression matrix was extracted from the training and validation datasets of RA and CAD . PCA analysis was performed on the new datasets using R packages FactoMineR and factoextra . SVM analysis was then conducted using functions from the R package e1071. Finally, the classification performance of the 25 core genes was visualized by ROC curves using pROC package.Fig.5 showed the 25 core genes distinguished the disease group from the normal group in both the RA and CAD training and validation datasets with AUCs above 0.8.

4.Lines 75 to 85 are not related to the introduction.

Dear Reviewer, 

Thank you for your valuable feedback. 

We have deleted the content of line75 to 85 in the introduction and revised as follows:

Potential targets and regulatory biological pathways were identified in RA and CAD datasets respectively using microarray bioinformatics technology in recent years, however, there is a dearth of studies that investigate the macrophages related inflammatory processes between RA and CAD. This research aims to fill this gap by employing GSEA analysis to identify common signaling pathways on RA and CAD datasets.WGCNA and ssGSEA will identify module genes linked to Fc-gamma receptor phagocytosis. Three hub genes will be extracted using STRING and Cytoscape, then assessing their predictive ability in both diseases through ROC analysis. GeneMANIA and the Cibersort algorithm will provide insights into immune infiltration and transcription factors associated with the hub genes. Notably, the involvement of PPARG and macrophages in RA and CAD will be highlighted. Understanding these processes could offer new insights into the pathogenesis shared by both diseases and potentially aid in the development of targeted therapies.

5.Most of the Figures are not clear eg, Fig 2 (A,B), Fig 4 (A,D,E), Fig 5 (A,B,C,D,F) , Fig 6 (E) and Fig 7 (A,B,C,D,E). The authors provide clearer Figures.

Thank you for your feedback on the clarity of our figures. We appreciate your attention to detail and would like to clarify the issue you mentioned.

We have compared the raw figures in TTF format and the figures of PDF versions, the latter indeed have lower resolution and pixel quality compared to the TTF versions. I think the review can try to accesse the raw figures in TTF format using the provided link in the top right corner of the PDF figures as marked in the following attached figures.We apologize for any inconvenience caused and appreciate your understanding. 

If you have any further questions or concerns, please do not hesitate to let us know. We appreciate your time and effort in reviewing our work.

 Pdf figure

Editor Comments

1.Fig 1. Need more explanation in legend.

Thank you for your feedback on Fig 1 legend. We have revised as follows: ＂Fig 1. Data analysis flowchart. Schematic flowchart of data aquirement, processing ,analysis and validation.＂

2.The text need to be improved as English Grammar and scientific frame.

Thank you for your suggestions. We have made revise by the editors and the proofreading certificate was uploaded in the supplementary materials.

3.Space and “,” should be checked in the whole text.

Thank you for your remind. We have checked the whole text to correct this mistake.

4.Some DE genes should be validated using qRT-PCR or other techniques. 

Thank you for your suggestions. We have performed qRT-PCR to validate TUBB2A, PPARG and FKBP5 mRNA expression in RA and CAD tissues and their mathced adjacent normal tissues.The results was demonstrated and in the results “GO and KEGG pathway analysis of 25 susceptible genes in RA and CAD”and showed in Fig.6G-H

5.Quality of figures should be improved and the text and details should be clear.

Thank you for your suggestions. We have assembled our figures with adobe illustration software and upload the figure again. If there is something unsatisfactory, please don’t hesitate to contact us.

Best wishes!

Yours sincerely!

---

## [Decision Letter · Decision Letter 1]

11 Jan 2024

PONE-D-23-28394R1Identification of PPARG as key gene to link coronary atherosclerosis disease and rheumatoid arthritis via microarray data analysisPLOS ONE

Dear Dr. zhang,

Thank you for submitting your manuscript to PLOS ONE. After careful consideration, we feel that it has merit but does not fully meet PLOS ONE’s publication criteria as it currently stands. Therefore, we invite you to submit a revised version of the manuscript that addresses the points raised during the review process.

We look forward to receiving your revised manuscript.

Kind regards,

Abozar Ghorbani, Ph.D

Academic Editor

PLOS ONE

Journal Requirements:

Reviewers' comments:

Reviewer's Responses to Questions

**Comments to the Author**

1. If the authors have adequately addressed your comments raised in a previous round of review and you feel that this manuscript is now acceptable for publication, you may indicate that here to bypass the “Comments to the Author” section, enter your conflict of interest statement in the “Confidential to Editor” section, and submit your "Accept" recommendation.

Reviewer #1: All comments have been addressed

Reviewer #3: All comments have been addressed

2. Is the manuscript technically sound, and do the data support the conclusions?

Reviewer #1: Yes

Reviewer #3: Yes

3. Has the statistical analysis been performed appropriately and rigorously? 

Reviewer #1: Yes

Reviewer #3: Yes

4. Have the authors made all data underlying the findings in their manuscript fully available?

Reviewer #1: Yes

Reviewer #3: Yes

5. Is the manuscript presented in an intelligible fashion and written in standard English?

Reviewer #1: Yes

Reviewer #3: Yes

6. Review Comments to the Author

Reviewer #1: 1. Briefly refer to the used data in the abstract of the method section

2. The reference genes should be specified in the real-time PCR section

3. What method has been used to analyze real-time PCR data?

Reviewer #3: The study is very interesting, as well as, the manuscript is improved very well, in my opinion, you can accept it. However, it is recommended to edit the following comments.

7. PLOS authors have the option to publish the peer review history of their article (what does this mean?). If published, this will include your full peer review and any attached files.

Reviewer #1: No

Reviewer #3: **Yes: **Fatemeh Yaghoobizadeh

---

## [Author Response · Author response to Decision Letter 1]

14 Jan 2024

Reviewer #1: The study is very interesting, as well as, the manuscript is improved very well, in my opinion, you can accept it. However, it is recommended to edit the following comments.

1.Briefly refer to the used data in the abstract of the method section

Response:Thank you for your comment. I have added the names of the datasets in the revised abstract of the method section as follows“The study utilized five raw datasets: GSE55235 , GSE55457 , GSE12021 for RA patients, and GSE42148 and GSE20680 for CAD patients in line 24-25”

2.The reference genes should be specified in the real-time PCR section

Response: Thank you for your comment. I have specified the reference genes as follows”GAPDH was used as reference gene” in line 165

3.What method has been used to analyze real-time PCR data?

Response: Thank you for your comment. I have mentioned the method usde to analyze real-time PCR data as follows”Gene expression was calculated according to the 2-ΔΔCT method .” in line 165-166.

Reviewer #3

The study is very interesting, as well as, the manuscript is improved very well, in my opinion, you can accept it. However, it is recommended to edit the following comments. 

1.Generally, the pronouns (such as I, we, us) are avoided to be written in a technical research article. In the regard, try to reframe the text, e.g., line 132, 173, 216, 277, 281, 303, etc. 

Response: Thank you for your comments. We have made revisions to the text accordingly. And the “revised manuscript with track changes” has been submitted for your review.

2.There is a blank page between the 154 and 156 lines. Please edit this error.

Response: Thank you for pointing out the error. We apologize for the oversight. We have now removed the blank page between lines 154 and 156 as shown in the “revised manuscript with track changes”. 

3.The diagrams quality is quite low. It is required to include high-resolution pictures for further comments.

Response: Thank you for your comments. We have already recombined the PDF images using Adobe Illustrator to generate vector graphics, exported them in TTF format with a PPI of 600dp, and re-uploaded them in the submitting system, The original images remain clear even when enlarged by 150%. I hope it will be OK.

4.Please review the numbers of figures. I think you mean “Figure 6” in line: 278, 282, 287, etc. and after the above-mentioned lines.

Response:Thank you for your comment. We have now reviewed the number of figures in the manuscript and made the necessary changes in lines 278, 282, 287, and other relevant lines, . The revised manuscript with track changes has been submitted for your review.

5.Please double check for spelling and English grammar errors (e.g., Space and “,”, and integral and correct time tense for whole document) in the manuscript.

Response: Thank you for your comment. We have now thoroughly reviewed the manuscript for spelling and English grammar errors. This includes checking for spaces, commas and the correct tense .The revised manuscript with track changes has been submitted for your review. 

6.In line 317, please clearly indicate the corresponding figure. 

Response: Thank you for bringing this to our attention. We have revised the manuscript according to your comment and clearly indicated the corresponding figure in line 317. We have made these changes in the revised manuscript with track changes. 

7.It seems lines 382-394 are redundant in the current position. It is recommended to insert the “Conclusion” section to state your total conclusion of the findings. 

Response: Thank you for your valuable suggestions. We have revised the manuscript accordingly. The redundant sections from lines 382-394 have been removed, and a "Conclusion" section has been inserted to provide a comprehensive summary of our findings. The revised manuscript with track changes has been submitted for your review. 

8.It is recommended to use an integral format of abbreviations and full form of terms, e.g. rheumatoid arthritis in line 437. 

Response: Thank you for your comment. We have made the necessary changes in the revised manuscript with track changes, Except for the first occurrence in the abstract and the introduction, where 'rheumatoid arthritis' is used in full form, the abbreviation 'RA' is used throughout the rest of the manuscript. Please review the revised manuscript for detail. 

9.Please double-check the author’s citation format in lines 435, 458, etc. 

Response: Thank you for your comment. I am not entirely sure about the issue with the citation format in lines 435 and 458, as I have checked and found that they correspond with the numbering of the references listed below. The citation format is consistent with the above ones as well and the published article in “plos one”. Could you please provide more specific details about the problem? Thank you for your patience.

10. Please double check the formats of all references. 

Response: Thank you for your comment. I have checked all the references and replaced reference 7 from Microsoft Azure platform with other reference. Some references have been supplemented with published articles from Plos One, including volumes (issues) and DOI links following the published article in Plos One.Please review the revised manuscript for detail.

---

## [Editor Report · Decision Letter 2]

2 Feb 2024

PONE-D-23-28394R2Identification of PPARG as key gene to link coronary atherosclerosis disease and rheumatoid arthritis via microarray data analysisPLOS ONE

Dear Dr. zhang,

Thank you for submitting your manuscript to PLOS ONE. After careful consideration, we feel that it has merit but does not fully meet PLOS ONE’s publication criteria as it currently stands. Therefore, we invite you to submit a revised version of the manuscript that addresses the points raised during the review process.

We look forward to receiving your revised manuscript.

Kind regards,

Abozar Ghorbani, Ph.D

Academic Editor

PLOS ONE

Journal Requirements:

**Additional Editor Comments:** Because of your request to correct your mistake, I am returning to you to submit the correct version.Authors comments: There are  some mistakes in the figrue order and loss of figure 9 as showed in the previewed pdf file

---

## [Author Response · Author response to Decision Letter 2]

5 Feb 2024

Reviewer #1: The study is very interesting, as well as, the manuscript is improved very well, in my opinion, you can accept it. However, it is recommended to edit the following comments.

1.Briefly refer to the used data in the abstract of the method section

Response:Thank you for your comment. I have added the names of the datasets in the revised abstract of the method section as follows“The study utilized five raw datasets: GSE55235 , GSE55457 , GSE12021 for RA patients, and GSE42148 and GSE20680 for CAD patients in line 24-25”

2.The reference genes should be specified in the real-time PCR section

Response: Thank you for your comment. I have specified the reference genes as follows”GAPDH was used as reference gene” in line 165

3.What method has been used to analyze real-time PCR data?

Response: Thank you for your comment. I have mentioned the method usde to analyze real-time PCR data as follows”Gene expression was calculated according to the 2-ΔΔCT method .” in line 165-166.

Reviewer #3

The study is very interesting, as well as, the manuscript is improved very well, in my opinion, you can accept it. However, it is recommended to edit the following comments. 

1.Generally, the pronouns (such as I, we, us) are avoided to be written in a technical research article. In the regard, try to reframe the text, e.g., line 132, 173, 216, 277, 281, 303, etc. 

Response: Thank you for your comments. We have made revisions to the text accordingly. And the “revised manuscript with track changes” has been submitted for your review.

2.There is a blank page between the 154 and 156 lines. Please edit this error.

Response: Thank you for pointing out the error. We apologize for the oversight. We have now removed the blank page between lines 154 and 156 as shown in the “revised manuscript with track changes”. 

3.The diagrams quality is quite low. It is required to include high-resolution pictures for further comments.

Response: Thank you for your comments. We have already recombined the raw PDF images using Adobe Illustrator to generate vector graphics, exported them in TTF format with a PPI of 300dpi, then checked and adjusted in the PACE system before re-uploaded them in the submitting system, The original images remain clear even when enlarged by 150%, which can be obtained from link on the the right corner of the pdf files. I hope it will be OK. I think figures in the pdf files may be compressed too much. 

4.Please review the numbers of figures. I think you mean “Figure 6” in line: 278, 282, 287, etc. and after the above-mentioned lines.

Response:Thank you for your comment. We have now reviewed the number of figures in the manuscript and made the necessary changes in lines 278, 282, 287, and other relevant lines, . The revised manuscript with track changes has been submitted for your review.

5.Please double check for spelling and English grammar errors (e.g., Space and “,”, and integral and correct time tense for whole document) in the manuscript.

Response: Thank you for your comment. We have now thoroughly reviewed the manuscript for spelling and English grammar errors. This includes checking for spaces, commas and the correct tense .The revised manuscript with track changes has been submitted for your review. 

6.In line 317, please clearly indicate the corresponding figure. 

Response: Thank you for bringing this to our attention. We have revised the manuscript according to your comment and clearly indicated the corresponding figure in line 317. We have made these changes in the revised manuscript with track changes. 

7.It seems lines 382-394 are redundant in the current position. It is recommended to insert the “Conclusion” section to state your total conclusion of the findings. 

Response: Thank you for your valuable suggestions. We have revised the manuscript accordingly. The redundant sections from lines 382-394 have been removed, and a "Conclusion" section has been inserted to provide a comprehensive summary of our findings. The revised manuscript with track changes has been submitted for your review. 

8.It is recommended to use an integral format of abbreviations and full form of terms, e.g. rheumatoid arthritis in line 437. 

Response: Thank you for your comment. We have made the necessary changes in the revised manuscript with track changes, Except for the first occurrence in the abstract and the introduction, where 'rheumatoid arthritis' is used in full form, the abbreviation 'RA' is used throughout the rest of the manuscript. Please review the revised manuscript for detail. 

9.Please double-check the author’s citation format in lines 435, 458, etc. 

Response: Thank you for your comment. I am not entirely sure about the issue with the citation format in lines 435 and 458, as I have checked and found that they correspond with the numbering of the references listed below. The citation format is consistent with the above ones as well and the published article in “plos one”. Could you please provide more specific details about the problem? Thank you for your patience.

10. Please double check the formats of all references. 

Response: Thank you for your comment. I have checked all the references and replaced reference 7 from Microsoft Azure platform with other reference. Some references have been supplemented with published articles from Plos One, including volumes (issues) and DOI links following the published article in Plos One.Please review the revised manuscript for detail. 

Journal requirements

Response: Thank you for your comment. I have checked all the references and replaced reference 7 from Microsoft Azure platform with other reference. Some references have been supplemented with published articles from Plos One, including volumes (issues) and DOI links following the published article in Plos One.Please review the revised manuscript for detail. None of the reference was restracted by now after throughly checked in the pubmed.

---

## [Decision Letter · Decision Letter 3]

21 Feb 2024

Identification of PPARG as key gene to link coronary atherosclerosis disease and rheumatoid arthritis via microarray data analysis

PONE-D-23-28394R3

Dear Dr. zhang,

We’re pleased to inform you that your manuscript has been judged scientifically suitable for publication and will be formally accepted for publication once it meets all outstanding technical requirements.

Kind regards,

Abozar Ghorbani, Ph.D

Academic Editor

PLOS ONE

Additional Editor Comments (optional):

Reviewers' comments:

Reviewer's Responses to Questions

**Comments to the Author**

1. If the authors have adequately addressed your comments raised in a previous round of review and you feel that this manuscript is now acceptable for publication, you may indicate that here to bypass the “Comments to the Author” section, enter your conflict of interest statement in the “Confidential to Editor” section, and submit your "Accept" recommendation.

Reviewer #1: All comments have been addressed

2. Is the manuscript technically sound, and do the data support the conclusions?

Reviewer #1: Yes

3. Has the statistical analysis been performed appropriately and rigorously? 

Reviewer #1: Yes

4. Have the authors made all data underlying the findings in their manuscript fully available?

Reviewer #1: Yes

5. Is the manuscript presented in an intelligible fashion and written in standard English?

Reviewer #1: Yes

6. Review Comments to the Author

Reviewer #1: (No Response)

7. PLOS authors have the option to publish the peer review history of their article (what does this mean?). If published, this will include your full peer review and any attached files.

Reviewer #1: No

---

## [Editor Report · Acceptance letter]

26 Mar 2024

PONE-D-23-28394R3 

PLOS ONE

Dear Dr. Zhang, 

I'm pleased to inform you that your manuscript has been deemed suitable for publication in PLOS ONE. Congratulations! Your manuscript is now being handed over to our production team.

Kind regards, 

on behalf of

Dr. Abozar Ghorbani 

Academic Editor

PLOS ONE